# Impact of economic indicators on rice production: A machine learning approach in Sri Lanka

Sherin Kularathne[1], Namal Rathnayake[2]*, Madhawa Herath[3], Upaka Rathnayake[4], Yukinobu Hoshino[5]

1 Faculty of graduate studies and research, Sri Lanka Institute of Information Technology, Malabe, Sri Lanka, 2 Graduate School of Engineering, The University of Tokyo, Bunkyo City, Tokyo, Japan, 3 Department of Mechanical Engineering, Faculty of Engineering, Sri Lanka Institute of Information Technology, Malabe, Sri Lanka, 4 Department of Civil Engineering and Construction, Faculty of Engineering and Design, Atlantic Technological University, Sligo, Ireland, 5 School of Systems Engineering, Kochi University of Technology, Tosayamada, Kami, Kochi, Japan

* namal@hydra.t.u-tokyo.ac.jp

**Data Availability Statement:** The data is stored in a public repository for reproducibility (https://www.kaggle.com/datasets/namalrathnayake1990/

## Abstract

Rice is a crucial crop in Sri Lanka, influencing both its agricultural and economic landscapes. This study delves into the complex interplay between economic indicators and rice production, aiming to uncover correlations and build prediction models using machine learning techniques. The dataset, spanning from 1960 to 2020, includes key economic variables such as GDP, inflation rate, manufacturing output, population, population growth rate, imports, arable land area, military expenditure, and rice production. The study's findings reveal the significant influence of economic factors on rice production in Sri Lanka. Machine learning models, including Linear Regression, Support Vector Machines, Ensemble methods, and Gaussian Process Regression, demonstrate strong predictive accuracy in forecasting rice production based on economic indicators. These results underscore the importance of economic indicators in shaping rice production outcomes and highlight the potential of machine learning in predicting agricultural trends. The study suggests avenues for future research, such as exploring regional variations and refining models based on ongoing data collection.

## Introduction

Rice, which provides nearly half the calories for half the world's population [1, 2] is a vital crop for Sri Lanka's agricultural and economic landscape, being the staple food, which determines food security and economic resiliency with more than 181kg per year consumption, marking the eighth greatest per capita consumption in the world [3]. Also, rice cultivation accounts for 34% of the total cultivated area, 1.8 million family farms, and 45% of all calories consumed by the average Sri Lankan [4].

The complex interplay of economic forces and rice production necessitates a thorough analysis and a nuanced knowledge of the multifaceted relationships that shape this dynamic.

economy-and-rice-production-sri-lanka-1960-2020).

**Funding:** This study was funded by the Japan Society for the Promotion of Science (JSPS) in the form of a Grants-in-Aid for Scientific Research (KAKENHI) grant to YH [22KK0160].

**Competing interests:** The authors have declared that no competing interests exist.

This study aims to contribute to this discussion by doing a thorough investigation and using machine learning approaches to uncover correlations and build prediction models.

The factors examined in this study are diversified facets of economic indicators, each with a unique impact on the intricate web of rice production. At the forefront is Gross Domestic Product (GDP), a vital indicator of the country's economic health that reflects and drives agricultural development. Along with GDP, the changing inflation is also examined. As Bhattarai mentioned [5], maintaining pricing stability is crucial for the economy's long-term growth and development. According to Atigala's findings on Sri Lanka [6], there is a negative relationship between inflation and economic growth in the short run as well. Furthermore, because growing inflation negatively impacts the economy, testing inflation's relationship with various economic factors is crucial for making proactive economic decisions. The strength of the manufacturing sector and the demographic underpinnings of population and growth can influence the setting in which rice production succeeds or fails, as they are some of the main economic indicators that have no proof of being tested out in the Sri Lankan context against rice production.

Moreover, the political instability in recent years has also been a motivating factor for conducting this study in Sri Lanka. Political instability can have significant implications for the economy, including the agricultural sector. By studying the relationship between economic indicators and rice production in Sri Lanka, the authors may aim to provide insights that can help mitigate the impact of political instability on rice cultivation and inform more stable and resilient agricultural policies. Additionally, the study seeks to understand how political factors interact with economic indicators to influence rice production, providing valuable insights for policymakers navigating the challenges of political uncertainty.

Furthermore, the study delves beyond the domestic domain, examining the impact of foreign trade using variables like imports. The quantity and quality of arable land, a finite resource crucial to agricultural sustainability, adds an environmental component to our inquiry. To reflect the broader economic backdrop, the study includes the influence of military expenditures, recognizing the interaction between military defence spending and its possible impact on the agricultural sector. Even though most studies in the literature [7–9] suggest that there is no significant consensus on the impact that military expenditure can have on the economic factors of a region, there is an absence of few studies that specifically focus on against rice production and in Sri Lankan context.

As this study navigates the complicated interdependence of these variables, machine learning can be used as a valuable instrument, capable of discovering patterns from vast data sets and providing insights into the deep relationships that influence economic factors and rice production. This study intends to find correlations while also providing a view into the future, allowing policymakers, agricultural stakeholders, and economists to make informed decisions.

Food production is a critical component of global agricultural systems, and understanding the factors influencing it has been the focus of extensive research. Over the years, numerous studies have investigated the relationship between food production and various methods, including statistical and machine-learning approaches [10, 11]. The following paragraphs aim to provide an overview of key research efforts in this area.

Climate change is one of the significant factors affecting food production, and several studies have explored its potential impact using statistical and machine learning methods Mulla et al. [12] and Rosenzweig et al. [13] simulated the effects of climate change on agriculture, highlighting the importance of temperature and precipitation. Lobell and Burke [14] examined the uncertainty in agricultural impacts of climate change, emphasizing the role of temperature relative to precipitation.

In addition to climate change, researchers have focused on understanding the patterns of crop yield growth and stagnation. Ray et al. [15] analyzed recent patterns of crop yield growth, while Liakos et al. [16] and Krishna et al. [17] reviewed machine learning approaches for crop yield prediction, highlighting the importance of these methods in agricultural research.

Machine learning has emerged as a powerful tool in agriculture, offering new opportunities for analyzing complex agricultural data. Satpathi et al. [18] conducted a comparative study of machine learning models for forecasting rice production, demonstrating the effectiveness of these models in predicting agricultural outcomes.

Vasilyevich [19] and Liakos et al. [16] explored the classification of agricultural data using machine learning algorithms, showcasing the potential of these methods in agricultural data analysis. Additionally, Furbank and Tester [20] discussed the role of phenomics technologies in relieving the phenotyping bottleneck in agriculture.

Machine learning algorithms have been applied to various aspects of agriculture, including agricultural data classification, flood prediction [21–25] and crop yield prediction [26, 27]. These studies demonstrate the versatility and effectiveness of machine learning in addressing agricultural challenges.

The study of economic indicators and their impact on rice production is a complex and dynamic field that requires sophisticated analytical tools. In recent years, fuzzy logic has emerged as a valuable approach for modeling complex systems with uncertain or imprecise data, making it particularly well-suited for economic predictions. Fuzzy logic allows for the representation of vague or ambiguous relationships between variables, which is often the case in economic systems where factors can be influenced by multiple variables with varying degrees of impact. For example, studies by Jagielska et al. [28] and Reghis et al. [29] have demonstrated the effectiveness of fuzzy logic in modeling economic systems with uncertain or incomplete information, providing insights into the decision-making processes that govern economic outcomes.

Furthermore, fuzzy logic has been applied to various aspects of economic forecasting, including stock market predictions, consumer behavior analysis, and macroeconomic modeling. For instance, studies by Ecer et al. [30] and Pant et al. [31] have used fuzzy logic to predict stock prices based on a range of economic indicators and market sentiment analysis. These studies have shown that fuzzy logic can capture the non-linear and uncertain nature of financial markets, leading to more accurate predictions compared to traditional statistical methods. In the context of rice production in Sri Lanka, fuzzy logic could offer a valuable tool for modeling the complex interactions between economic factors and agricultural outcomes, providing insights that can enhance the resilience and sustainability of rice cultivation in the face of economic uncertainties [32, 33].

Hybrid fuzzy systems, such as the Adaptive Neuro-Fuzzy Inference System (ANFIS) and its variations [34, 35], have gained popularity for their ability to combine the strengths of fuzzy logic and neural networks in modeling complex economic systems. ANFIS integrates fuzzy logic principles with neural network learning algorithms, allowing for the automatic learning of fuzzy rules from data. This approach has been successfully applied in various economic forecasting tasks, including time series prediction, financial market analysis, and macroeconomic modeling. For example, studies by Jang et al. [36], and Boyacioglu et al. [37] have demonstrated the effectiveness of ANFIS in predicting stock prices and exchange rates, showcasing its ability to capture the non-linear and uncertain relationships inherent in financial markets. In the context of rice production in Sri Lanka, the use of hybrid fuzzy systems like ANFIS could offer a powerful tool for modeling the complex interactions between economic indicators and agricultural outcomes, providing accurate predictions and valuable insights for decision-making.

In conclusion, the relationship between food production and statistical/machine learning methods has been the subject of extensive research. Studies have highlighted the importance of these methods in understanding the complex dynamics of agricultural systems, predicting crop yields, and addressing challenges such as climate change impacts. As technology continues to advance, the application of statistical and machine learning methods in agriculture is expected to play an increasingly significant role in shaping the future of food production.

This research aims to go beyond a surface-level understanding of economic factors and rice production in Sri Lanka. By incorporating machine learning approaches, the research hopes to contribute to a more sophisticated understanding of the connected threads that connect the country's economic growth to the health of its agricultural backbone. Through such insights, this research can pave the road for evidence-based policies and initiatives that promote a resilient future for Sri Lankan rice production in the face of a dynamic economic landscape.

The comprehensive investigation presented in the above paragraphs points out the following flaws of past studies.

1. Limited Scope: Past studies may have focused on a narrow set of economic indicators or environmental factors, overlooking the complexity and interdependence of various factors that influence rice production.

2. Lack of Local Context: Some studies may have lacked a specific focus on the Sri Lankan context, potentially leading to findings that are not directly applicable to the local agricultural landscape.

3. Static Analysis: Previous research might have relied on static analysis methods without leveraging advanced machine learning techniques, which could limit the depth of insights into the dynamic relationships between economic factors and rice production.

Hence, this study focuses on achieving the following objectives.

1. Comprehensive Analysis: This research aims to provide a thorough investigation of the multifaceted relationships between economic indicators and rice production in Sri Lanka, offering a more comprehensive understanding of the factors at play.

2. Integration of Machine Learning: By incorporating machine learning approaches, the study seeks to uncover correlations and build prediction models that can provide insights into the complex dynamics of economic forces and rice cultivation.

3. Potential Policy Implications: The findings of this research could inform evidence-based policies and initiatives that promote a resilient future for Sri Lankan rice production, highlighting the practical applications of the study's insights for policymakers and agricultural stakeholders.

This study presents a novel and comprehensive analysis of the multifaceted relationships between economic indicators and rice production in Sri Lanka, integrating machine learning approaches to uncover correlations and build prediction models, with findings directly applicable to the local agricultural landscape and potential implications for evidence-based policies and initiatives.

The findings of this study contribute significantly to addressing the challenges identified in the introduction by offering a comprehensive analysis of the complex relationships between economic indicators and rice production in Sri Lanka. By integrating machine learning approaches, the study provides a deeper understanding of these relationships, which can inform evidence-based policies and initiatives to promote a resilient future for Sri Lankan rice production. Additionally, the focus on the local context ensures that the findings are directly

applicable to the specific challenges faced by the country's agricultural sector, bridging the gap in the literature and providing actionable insights for policymakers and agricultural stakeholders.

## Methodology

Fig 1 represents the overall flow of the study. The study follows a structured flow beginning with data collection, where comprehensive datasets spanning from 1960 to 2020 are gathered from various sources. The next step involves data preparation, where the collected data is cleansed, integrated, and transformed to ensure consistency and compatibility for analysis. This leads to the data exploration phase, wherein the distribution and characteristics of the variables are carefully examined to gain insights into their behavior over time.

$$RiceProduction = function \begin{cases} GDP(\$B) \\ Inflation(\%) \\ Manufacturing(\$B) \\ Population \\ GrowthRate(\%) \\ Imports(\$B) \\ ArableLand \\ Military(\$B) \end{cases} \quad (1)$$

Following this, correlation analysis is conducted to identify the variables that exhibit strong correlations with rice production, shedding light on potential influential factors. Eq 1 summarizes the assumed relationship between economic factors and rice production.

Subsequently, traditional and modern state-of-the-art machine learning regression models are trained using the prepared dataset to predict rice production based on the identified factors.

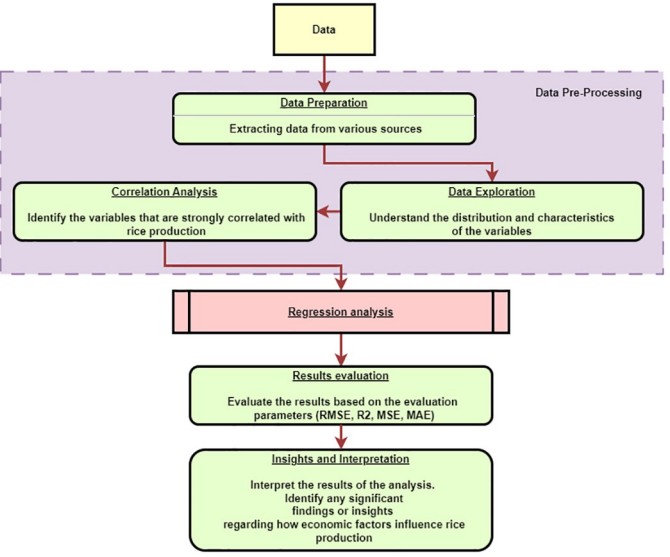

**Fig 1. Overall flow of the study.**

The results of the trained models are evaluated using standard evaluation parameters such as Root Mean Squared Error (RMSE), R-squared (R2), Mean Squared Error (MSE), and Mean Absolute Error (MAE) to assess their predictive performance. Finally, the study concludes with insights and interpretations drawn from the results, highlighting the significant economic factors that influence rice production in Sri Lanka. This comprehensive approach aims to provide a general understanding of the relationship between economic variables and rice production, contributing to the region's broader understanding of agricultural dynamics.

## Data description and preprocessing

The dataset spans from 1960 to 2020 and includes various economic indicators and rice production data from Sri Lanka. These indicators consist of the country's GDP in billions of dollars, inflation rate as a percentage, manufacturing output in billions of dollars, total population count, population growth rate as a percentage, imports in billions of dollars, arable land area in square kilometres, military expenditure in billions of dollars, and rice production in metric tons. The dataset's range of values varies across different years and indicators. For instance, the GDP ranges from a minimum of 1.24 billion dollars in 1963 to a maximum of 94.49 billion dollars in 2018, while the inflation rate fluctuates between -0.02% and 0.26%. Similarly, other variables like manufacturing output, population, imports, arable land, military expenditure, and rice production also exhibit varying ranges over the years. In addition to the dataset containing economic indicators and rice production data from Sri Lanka between 1960 and 2020, further statistical information is available for each variable. This includes the count, mean, standard deviation (std), minimum (min), 25th percentile (25%), 50th percentile (50% or median), 75th percentile (75%), and maximum (max) values for each variable. A descriptive analysis of the dataset is presented in Table 1.

To prepare the dataset for analysis, several preprocessing steps can be undertaken. This includes checking for missing or inconsistent values and handling them appropriately, such as through imputation or removal of outliers. Additionally, normalizing or standardizing numeric variables may be necessary to ensure that they are on a similar scale, especially if using machine learning algorithms that are sensitive to scale differences. Feature engineering could also enhance the dataset's predictive power by creating or transforming new features. Furthermore, throughout this study, the dataset was divided into training and testing sets using a consistent 7:3 ratio. The same train and test dataset was used consistently for all analyses and experiments.

**Table 1. Summary of statistical measures for economic and agricultural variables in Sri Lanka (1960-2020).**

| Variables | count | mean | std | min | 25% | 50% | 75% | max |
|---|---|---|---|---|---|---|---|---|
| Year | 61 | 1990 | 17.75 | 1960 | 1975 | 1990 | 2005 | 2020 |
| GDP ($B) | 61 | 23 | 29.47 | 1.24 | 3.36 | 8.03 | 24.41 | 94.49 |
| Inflation (%) | 61 | 0 | 0.06 | -0.02 | 0.04 | 0.08 | 0.12 | 0.26 |
| Manufacturing ($B) | 61 | 4 | 4.95 | 0.2 | 0.61 | 1.08 | 4.76 | 14.39 |
| Population | 61 | $1.66 \times 10^7$ | $3.64 \times 10^6$ | $9.78 \times 10^6$ | $1.37 \times 10^7$ | $1.72 \times 10^7$ | $1.97 \times 10^7$ | $2.17 \times 10^7$ |
| Growth Rate (%) | 61 | 0 | 0.01 | 0 | 0.01 | 0.01 | 0.02 | 0.02 |
| Imports ($B) | 61 | 7 | 7.64 | 0.34 | 1.13 | 3.06 | 10.07 | 26.8 |
| Arable Land | 61 | $9.58 \times 10^5$ | $1.99 \times 10^5$ | $5.77 \times 10^5$ | $8.05 \times 10^5$ $1.97 \times 10^7$ | $9.00 \times 10^5$ | $1.00 \times 10^6$ | $1.37 \times 10^6$ |
| Military($B) | 61 | 1 | 0.66 | 0.01 | 0.03 | 0.28 | 0.82 | 2.06 |
| Rice Production (Mt.) | 61 | 2311 | 883.53 | 758 | 1605 | 2389 | 2860 | 4301 |

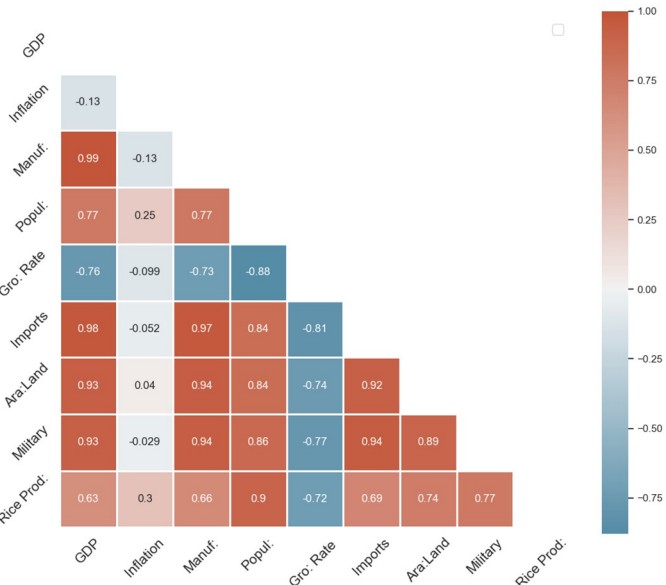

**Fig 2. The correlation coefficients of selected variables with 'Rice Production (Mt.)'.**

## Correlation analysis

The correlation analysis reveals interesting insights into the relationships between various factors and rice production ('Rice Production (Mt.)'). The highest positive correlation between rice production and the year indicates a potential trend of increasing rice production over time. Additionally, strong positive correlations are found between rice production and economic indicators such as GDP ('Gross Domestic Product'), manufacturing output, population, imports, arable land, and military expenditure. These findings suggest that economic growth, population dynamics, and resource availability, as represented by arable land and imports, may significantly influence rice production. Conversely, a notable negative correlation is observed between rice production and the growth rate, indicating a potential inverse relationship between economic growth and rice production (Refer to Fig 2).

## Machine learning models

This study employs a range of machine learning models, spanning traditional algorithms like Linear Regression, Decision Trees, and Support Vector Regression, which are renowned for their interpretability and efficiency. On the cutting edge, modern techniques like Neural Networks and Gaussian Process Regression harness the power of deep learning and ensemble methods to capture intricate data relationships. This section comprehensively explores these algorithms, each contributing its unique strengths to accurate prediction.

**Traditional ML models.** *Linear regression (Linear).* Linear regression is a basic and widely used statistical method for modeling the relationship between a dependent variable and one or more independent variables. It assumes a linear relationship between the input variables and the output. The pseudo-code of the algorithm is shown in 1 and tuned hyper parameters are shown in Table 2.

**Algorithm 1** Linear Regression

```
1: procedure LINEARREGRESSION(X, y)
   (X represents the feature matrix, y represents the target variable)
```

**Table 2. Tuned hyperparameters for linear regression.**

| Hyperparameter | Value |
| --- | --- |
| fit_intercept | True |
| normalize | True |

**Table 3. Hyperparameters for robust linear regression.**

| Hyperparameter | Value |
| --- | --- |
| delta | 1.0 |
| num_iterations | 100 |

**Table 4. Hyperparameters for decision trees.**

| Tree Complexity | Hyperparameters |
| --- | --- |
| Fine | Max Depth: 10, Min Samples Split: 2 |
| Medium | Max Depth: 5, Min Samples Split: 5 |

```
2:      Initialize the coefficients vector β
3:      Add a column of ones to the feature matrix X (for the intercept
        term)
4:      Compute the least squares solution: β = (XᵀX)⁻¹Xᵀy
5:      return β
6: end procedure
```

*Linear regression (Robust)*. Robust linear regression is a variation of linear regression that is less sensitive to outliers in the data. It uses robust estimation techniques to minimize the impact of outliers on the model's parameters. The pseudo-code of the algorithm is shown in 2 and tuned hyper parameters are shown in Table 3.

**Algorithm 2** Robust Linear Regression (Huber Regression)

```
1: procedure RobustLinearRegression(X, y, δ, num_iterations)
   (X represents the feature matrix, y represents the target variable,
   and δ represents the threshold for the Huber loss function)
2:      Initialize the coefficients vector β
3:      Add a column of ones to the feature matrix X (for the intercept
        term)
4:      for i = 1 to num_iterations do
5:        Compute the residuals: r = y − Xβ
6:        Compute the weights using Huber loss function: W = diag(1, ...,
          1) if |r| ≤ δ, otherwise W = diag(δ/|r|)
7:        Update the coefficients using weighted least squares: β =
          (XᵀWX)⁻¹XᵀWy
8:      end for
9:      return β
10: end procedure
```

*Tree (Fine, Medium)*. These refer to decision tree models with different levels of complexity. Decision trees partition the input space into regions and assign a constant value to each region, making them easy to interpret. The pseudo-code of the algorithm is shown in 3 and tuned hyperparameters are shown in Table 4.

**Algorithm 3** Decision Tree

```
1: procedure DecisionTree(X, y, max_depth)
```

```
2:    if max_depth = Fine then
3:      Call RecursiveTree(X, y, max_depth, Fine)
4:    else if max_depth = Medium then
5:      Call RecursiveTree(X, y, max_depth, Medium)
6:    end if
7: end procedure
8: procedure RECURSIVETREE(X, y, max_depth, complexity)
9:    if stopping_criterion_met then
10:     Create a leaf node with the majority class in y
11:   else if complexity = Fine then
12:     Select the best split attribute and value
13:     Split the data into two subsets based on the selected split
14:     Call RecursiveTree(X_left, y_left, max_depth−1, Fine)
15:     Call RecursiveTree(X_right, y_right, max_depth−1, Fine)
16:   else if complexity = Medium then
17:     Select the best split attribute and value using a faster
          heuristic
18:     Split the data into two subsets based on the selected split
19:     Call RecursiveTree(X_left, y_left, max_depth−1, Medium)
20:     Call RecursiveTree(X_right, y_right, max_depth−1, Medium)
21:   end if
22: end procedure
```

*SVM (Linear, Quadratic, Cubic, Fine Gaussian, Medium Gaussian, Coarse Gaussian).* Support Vector Machines (SVMs) are a class of supervised learning algorithms used for classification and regression tasks. They work by finding the hyperplane that best separates the classes in the input space. The variations you've listed refer to different types of kernel functions used in SVMs, which determine the shape of the decision boundary. The pseudo-code of the algorithm is shown in 4 and tuned hyperparameters are shown in Table 5.

**Algorithm 4** Support Vector Machine

```
1: procedure SVM(X, y, kernel, complexity)
2:    if kernel = Linear then
3:      Call LinearSVM(X, y, complexity)
4:    else if kernel = Quadratic then
5:      Call PolynomialSVM(X, y, 2, complexity)
6:    else if kernel = Cubic then
7:      Call PolynomialSVM(X, y, 3, complexity)
8:    else if kernel = Fine Gaussian then
9:      Call GaussianSVM(X, y, Fine)
10:   else if kernel = Medium Gaussian then
11:     Call GaussianSVM(X, y, Medium)
12:   else if kernel = Coarse Gaussian then
13:     Call GaussianSVM(X, y, Coarse)
14:   end if
15: end procedure
```

**Table 5. Hyperparameters for support vector machines.**

| Kernel Type | Hyperparameters |
| --- | --- |
| Linear | C: 1.0 |
| Quadratic | C: 1.0, Degree: 2 |
| Cubic | C: 1.0, Degree: 3 |
| Fine Gaussian | C: 1.0, Gamma: 0.001 |
| Medium Gaussian | C: 1.0, Gamma: 0.01 |
| Coarse Gaussian | C: 1.0, Gamma: 0.1 |

```
16: procedure LINEARSVM(X, y, complexity)
17:    Train SVM with linear kernel using optimization algorithm with
       complexity setting
18: end procedure
19: procedure POLYNOMIALSVM(X, y, degree, complexity)
20:    Train SVM with polynomial kernel of degree degree using optimi-
       zation algorithm with complexity setting
21: end procedure
22: procedure GAUSSIANSVM(X, y, granularity)
23:    if granularity = Fine then
24:       Train SVM with fine Gaussian kernel using optimization
          algorithm
25:    end if granularity = Medium then
26:       Train SVM with medium Gaussian kernel using optimization
          algorithm
27:    end if granularity = Coarse then
28:       Train SVM with coarse Gaussian kernel using optimization
          algorithm
29:    end if
30: end procedure
```

*Ensemble (Boosted, Bagged).* Ensemble methods combine multiple models to improve predictive performance. Sequentially boosting train models, with each subsequent model focusing on the errors of the previous ones. Bagging trains multiple models independently and combines their predictions through averaging or voting. The pseudo-codes of the algorithm are shown in 5 and 6, and tuned hyperparameters are shown in Table 6.

**Algorithm 5** Boosting

```
1: procedure BOOSTING(X, y, base_learner, num_learners)
2:    Initialize the weights w_i = 1/n, where n is the number of samples
3:    Initialize an empty list of weak learners: h_1, h_2, ..., h_num_learners
4:    for t = 1 to num_learners do
5:       Train a weak learner h_t using weighted samples X and y with
         weights w
6:       Compute the error ε_t of h_t on the weighted samples
7:       Compute the weight α_t = 1/2 log((1-ε_t)/(ε_t))
8:       Update the weights: w_i ← w_i · exp(-α_t y_i h_t(x_i)), where x_i is the
         i-th sample
9:       Normalize the weights: w ← w / (∑_{i=1}^{n} w_i)
10:      Add h_t to the list of weak learners
11:   end for
12:   return h_1, h_2, ..., h_num_learners
13: end procedure
```

**Algorithm 6** Bagging

```
1: procedure BAGGING(X, y, base_learner, num_learners)
2:    Initialize an empty list of weak learners: h_1, h_2, ..., h_num_learners
3:    for t = 1 to num_learners do
4:       Sample n examples from X with replacement
5:       Train a weak learner h_t on the sampled data
```

**Table 6. Hyperparameters for ensemble learning.**

| Ensemble Method | Hyperparameters |
|---|---|
| Boosted | Number of Estimators: 100, Learning Rate: 0.1 |
| Bagged | Number of Estimators: 100, Bootstrap Samples: True |

```
6:     Add h_t to the list of weak learners
7:   end for
8:   return h_1, h_2, ..., h_num_learners
9: end procedure
```

**Modern ML models.** *Gaussian Process Regression (Squared Exp, Matern 5/2, Exponential, Rational Quadratic Exp)*. GPR is a non-parametric Bayesian approach to regression. It models the target variable as a Gaussian process, which is characterized by a mean function and a covariance function (kernel). The variations that were captured refer to different kernel functions used in GPR, which capture different types of smoothness in the data. The pseudo-code of the algorithm is shown in 7 and tuned hyperparameters are shown in Table 7.

**Algorithm 7** Gaussian Process Regression

```
1: procedure GAUSSIANPROCESSREGRESSION(X, y, kernel)
2:   Compute the kernel matrix K using the selected kernel function
     and input data X
3:   Add a small noise term σ²I to the kernel matrix K for numerical
     stability
4:   Compute the Cholesky decomposition of the kernel matrix: L = Cho-
     lesky(K + σ²I)
5:   Compute the alpha vector: α = (L^T)^{-1}(L^{-1}y)
6:   return mean function and covariance function determined by K
     and α
  (The kernel matrix K is computed based on the chosen kernel function
and input data X. To ensure numerical stability, a small noise term σ²I
is added to the kernel matrix before performing the Cholesky
decomposition.)
7: end procedure
```

These models represent a diverse set of machine learning tools, each with its own strengths and weaknesses. When choosing a model for a specific task, it's important to consider factors such as the data's nature, the problem's complexity, and the interpretability of the model's results. These algorithms represent a diverse toolkit, each offering unique advantages for modeling the complex relationships within the economic factors affecting rice production.

## Evaluation parameters

This section explores key performance metrics employed to assess the accuracy and reliability of the developed wind power prediction models. Root Mean Squared Error (RMSE), Mean Absolute Error (MAE), Mean Squared Error (MSE), and Coefficient of Determination ($R^2$) are discussed in detail, providing insights into the models' predictive capabilities. The analysis of these metrics enhances our understanding of the model's accuracy, highlighting their strengths and areas for improvement.

**Root Mean Squared Error (RMSE).** RMSE is a widely used metric that measures the average magnitude of the errors between predicted ($y_i$) and observed ($\bar{y}_i$) values (Eq 2) [38]. It comprehensively assesses the model's accuracy, where lower values indicate better

**Table 7. Hyperparameters for gaussian process regression.**

| Kernel Function | Hyperparameters |
|---|---|
| Squared Exponential | Length Scale: 1.0, Variance: 0.1 |
| Matern 5/2 | Length Scale: 0.5, Variance: 0.2 |
| Exponential | Length Scale: 0.8, Variance: 0.15 |
| Rational Quadratic Exp | Length Scale: 1.2, Scale Mixture: 0.3, Variance: 0.1 |

performance.

$$RMSE = \sqrt{\frac{1}{N}\sum_{i=1}^{N}(y_i - \bar{y}_i)^2} \tag{2}$$

Where $N$ is the total number of observations.

**Mean Absolute Error (MAE).** MAE quantifies the average absolute difference between predicted ($\hat{y}_i$) and observed ($y_i$) values, providing a measure of the model's accuracy without considering the direction of the errors (Eq 3) [39].

$$MAE = \frac{1}{N}\sum_{i=1}^{N}|y_i - \hat{y}_i| \tag{3}$$

Where N is the total number of observations.

**Mean Squared Error (MSE).** MSE measures the average of the squares of the errors between predicted ($\hat{y}_i$) and observed ($y_i$) values, providing a measure of the model's accuracy that emphasizes larger errors than MAE (Eq 4) [40].

$$MSE = \frac{1}{N}\sum_{i=1}^{N}(y_i - \hat{y}_i)^2 \tag{4}$$

**Coefficient of Determination ($R^2$).** The Coefficient of Determination, often denoted as $R^2$, assesses the proportion of the variance in the dependent variable (wind power generation) that is predictable from the independent variables (features). It ranges between 0 and 1, where higher values indicate better explanatory power (Eq 5) [33].

$$R^2 = 1 - \frac{\sum_{i=1}^{N}(y_i - \hat{y}_i)^2}{\sum_{i=1}^{N}(y_i - \bar{y}_i)^2} \tag{5}$$

Where $N$ is the total number of observations. $y_i$ is the observed value. $\hat{y}_i$ is the predicted value. $\bar{y}_i$ is the mean of the observed values.

These performance indexes collectively evaluate the model's predictive accuracy, highlighting different aspects of the prediction errors.

## Results and discussion

Figs 3 and 4 depict the top eight test results of prediction and actual values using a scatter plot. These results highlight the comparative performance of various machine learning algorithms in predicting the target variable. The evaluated algorithms include Gaussian Process Regression (GPR) with different kernels (Exponential, Squared Exponential, and Matern), Support Vector Machine (SVM) with a linear kernel, Linear Regression (both linear and robust), Ensemble bagging, and Decision Tree with coarse branching.

Table 8 presents the comprehensive results of the study. The GPR models consistently exhibit strong performance, with all three variants achieving high R2 values of 0.94. This indicates a high degree of explained variance in the predictions, supported by low RMSE, MSE, and MAE values, indicating accurate prediction of the target variable.

Similarly, the SVM with a linear kernel demonstrates promising results, achieving an R2 value of 0.94 and competitive RMSE, MSE, and MAE values. The linear regression models also exhibit strong predictive capabilities, with R2 values of 0.92 and low error metrics.

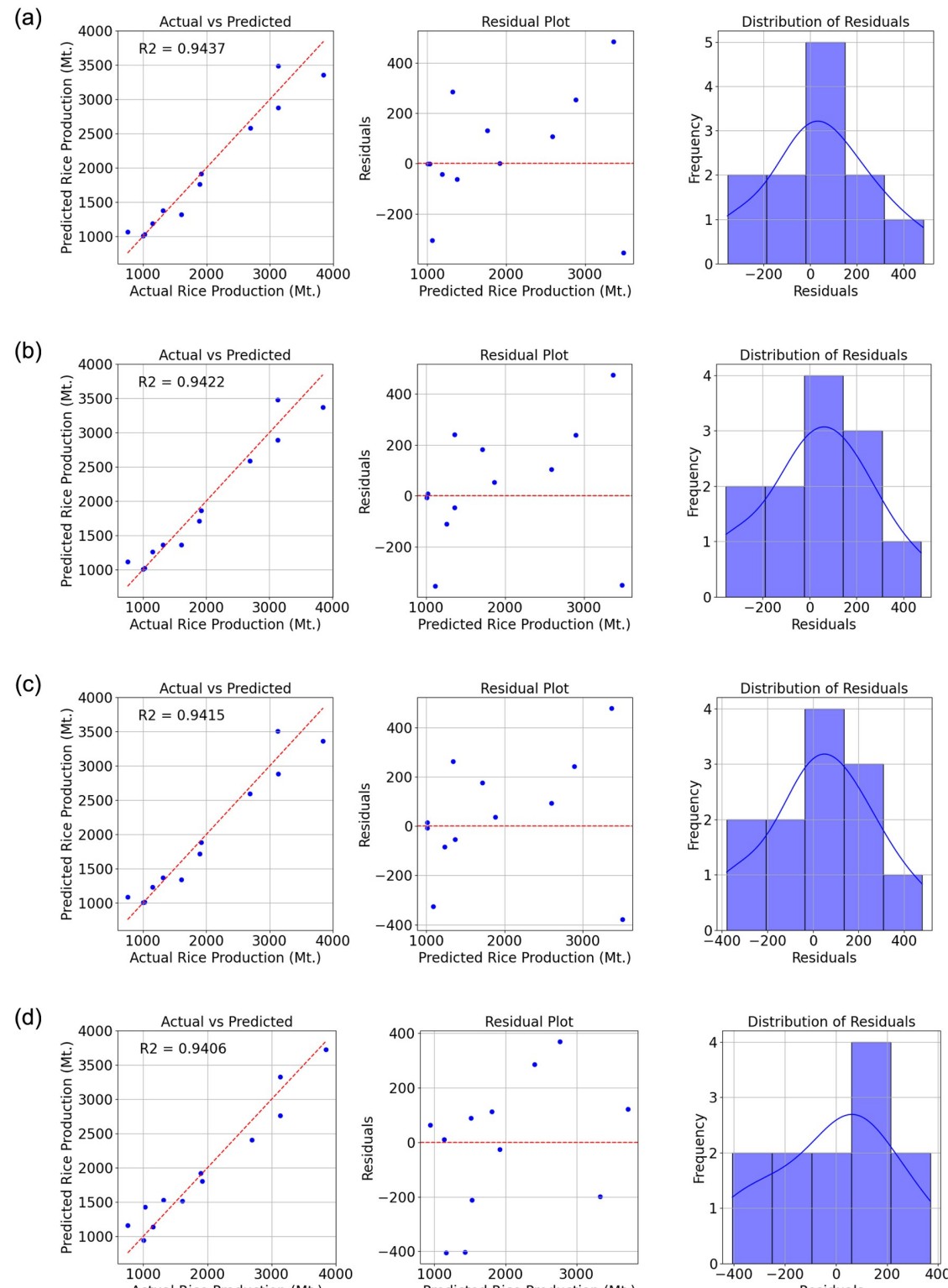

**Fig 3. Comparison of actual and predicted values, residual plot, and residual distribution.** (a) Gaussian Process Regression (GPR)—Exponential (b) Gaussian Process Regression (GPR)—SquaredExponential (c) Gaussian Process Regression (GPR)—Matern (nu = 5/2) and (d) Support Vector Machine (SVM)—Linear.

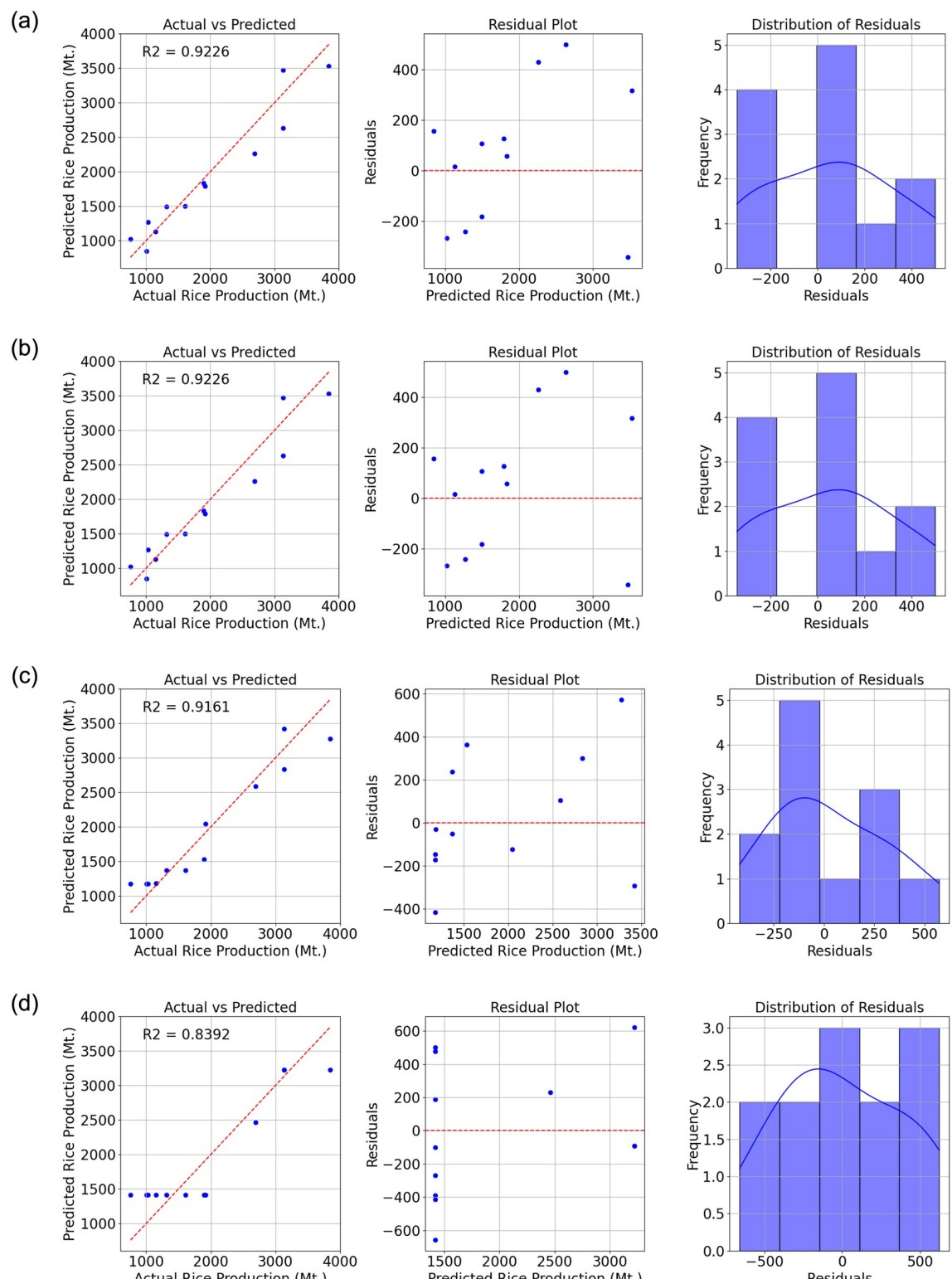

**Fig 4. Comparison of actual and predicted values, residual plot, and residual distribution.** (a) Linear Regression—Linear (b) Linear Regression—Robust (c) Ensemble bagging and (d) Decision tree—Coarse.

**Table 8. Performance metrics (R2, RMSE, MSE, MAE) for eight machine learning algorithms in predicting economic factors related to rice production.**

| Algorithm | R2 | RMSE | MSE | MAE |
|---|---|---|---|---|
| Gaussian Process Regression (GPR) Exponential | 0.9437 | 230 | 52886 | 169 |
| Gaussian Process Regression (GPR) SquaredExponential | 0.9422 | 233 | 54332 | 181 |
| Gaussian Process Regression (GPR) Matern (nu = 5/2) | 0.9415 | 234 | 54989 | 180 |
| Support Vector Machine (SVM) Linear | 0.9406 | 236 | 55830 | 191 |
| Linear Regression Linear | 0.9226 | 270 | 72698 | 229 |
| Linear Regression Robust | 0.9226 | 270 | 72698 | 229 |
| Ensemble bagging | 0.9161 | 281 | 78819 | 235 |
| Decision tree Coarse | 0.8392 | 389 | 151046 | 336 |

The Ensemble bagging model performs adequately with an R2 value of 0.92, indicating good predictive performance, although slightly lower than the GPR, SVM, and Linear Regression models. In contrast, the Decision Tree model with coarse branching demonstrates the weakest performance among the algorithms considered, with an R2 value of 0.84 and higher error metrics, indicating a less accurate prediction of the target variable compared to other models.

The study compared the responses of the testing predictions generated by eight different algorithms and presented the results in Figs 5 and 6.

In addition to the performance comparison, it is essential to consider the broader implications of the study. The findings of this study can be compared with similar studies in the literature to provide a more comprehensive understanding of the performance of different machine learning algorithms in predicting economic factors related to rice production. Furthermore, the study provides valuable insights into the advantages of using GPR, SVM with a linear kernel, and Linear Regression models for similar prediction tasks.

The study's comprehensive analysis of the relationship between economic indicators and rice production in Sri Lanka offers valuable insights that can inform evidence-based policy-making and agricultural strategies. By uncovering the complex interplay of factors influencing rice production, the study provides a nuanced understanding of the challenges and opportunities facing the agricultural sector in Sri Lanka.

One practical implication of the study is its potential to guide policymakers in developing strategies to enhance rice production and agricultural sustainability. By identifying key economic indicators that impact rice cultivation, policymakers can tailor interventions to address specific challenges and capitalize on opportunities for growth. For example, the study's findings suggest that focusing on boosting the manufacturing sector and maintaining price stability could positively influence rice production. Similarly, strategies to improve arable land quality and manage inflation could lead to more resilient agricultural systems.

Furthermore, the study's use of machine learning approaches adds a new dimension to its practical implications. By demonstrating the effectiveness of these techniques in predicting rice production outcomes, the study highlights the potential for integrating advanced technologies into agricultural decision-making. This could lead to more accurate and timely predictions, allowing farmers and policymakers to respond proactively to changing conditions.

Moreover, the study opens up avenues for future research. Further investigations could explore the application of other machine learning algorithms or ensemble methods to improve predictive performance. Additionally, incorporating more features or datasets could enhance the accuracy of the predictions. This study lays the foundation for future research in the field of predicting economic factors related to rice production using machine learning algorithms.

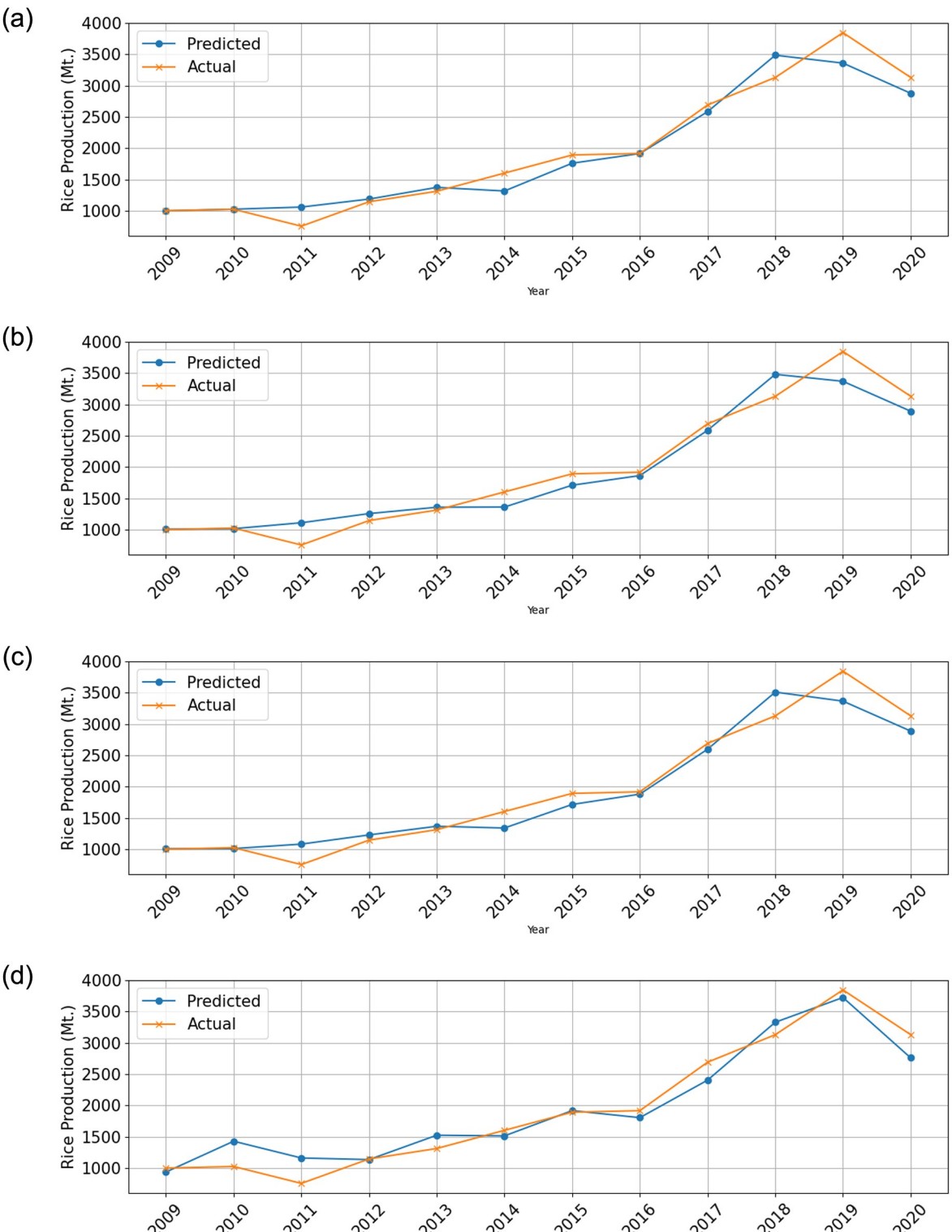

**Fig 5. Response plot showing predicted and actual values over time.** (a) Gaussian Process Regression (GPR)—Exponential (b) Gaussian Process Regression (GPR)—SquaredExponential (c) Gaussian Process Regression (GPR)—Matern (nu = 5/2) and (d) Support Vector Machine (SVM)—Linear.

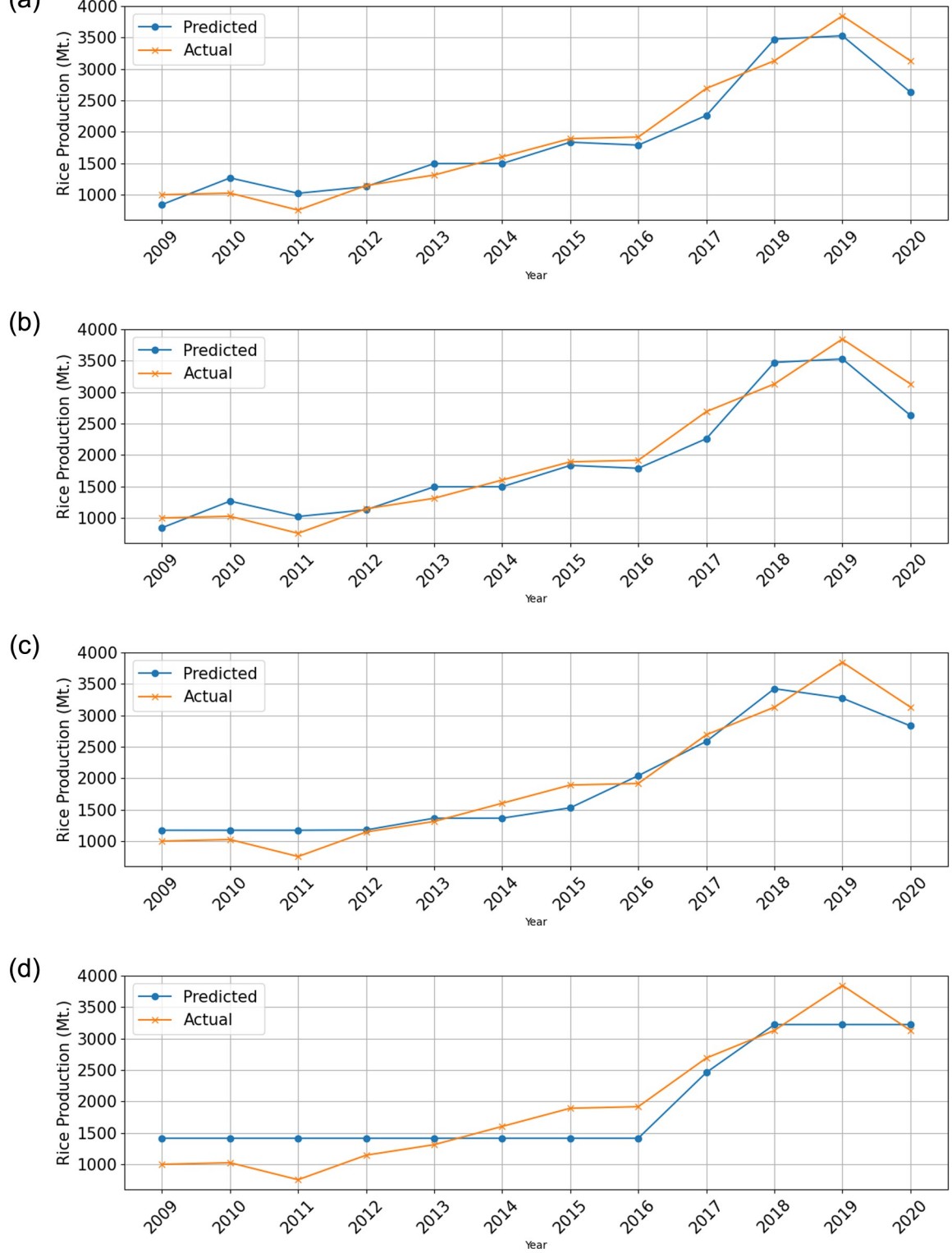

**Fig 6. Response plot showing predicted and actual values over time.** (a) Linear Regression—Linear (b) Linear Regression—Robust (c) Ensemble bagging and (d) Decision tree—Coarse.

In summary, this study provides a detailed comparison of different machine learning algorithms' performance in predicting economic factors related to rice production. The results offer valuable insights for researchers, policymakers, and stakeholders interested in selecting the most suitable machine learning algorithm for similar prediction tasks.

## Conclusion

In conclusion, this study provides a novel and comprehensive analysis of the multifaceted relationships between economic indicators and rice production in Sri Lanka, integrating machine learning approaches to uncover correlations and build prediction models. The findings of this research offer practical implications for policymakers and agricultural stakeholders, providing insights that can inform evidence-based policies and initiatives to promote a resilient future for Sri Lankan rice production. The study's focus on the local context ensures that the findings are directly applicable to the specific challenges faced by the country's agricultural sector, bridging the gap in the literature and providing actionable insights for decision-makers. Furthermore, the transferability of the study's results to other locations facing similar challenges of political instability and reliance on rice as a staple food highlights the broader impact and usefulness of this work.

The transferability of the study's results to other locations can be significant, particularly in regions facing similar challenges of political instability and where rice is a staple food. The findings can offer valuable insights into how economic indicators impact rice production, which can be applied to other countries experiencing political turmoil. Additionally, since many countries rely on rice as a staple food, the study's results can be useful for policymakers and stakeholders globally, providing a framework for understanding the complex interplay between economic factors and rice cultivation. By considering the specific context of Sri Lanka's political instability and the widespread use of rice as a staple, the study's findings can serve as a blueprint for enhancing agricultural sustainability and economic resilience in other regions facing similar circumstances.

Furthermore, this study paves the way for future research by suggesting several potential avenues. Subsequent investigations could delve into employing alternative machine learning algorithms or ensemble methods to enhance predictive accuracy. Additionally, the incorporation of additional features or datasets could further refine the predictive models. Overall, this study contributes significantly to addressing the challenges identified in the introduction and lays the foundation for future research on using machine learning algorithms to predict economic factors related to rice production.

## Author Contributions

**Conceptualization:** Sherin Kularathne, Namal Rathnayake.

**Data curation:** Namal Rathnayake, Upaka Rathnayake.

**Methodology:** Sherin Kularathne, Namal Rathnayake.

**Software:** Namal Rathnayake.

**Supervision:** Madhawa Herath, Upaka Rathnayake, Yukinobu Hoshino.

**Validation:** Madhawa Herath, Upaka Rathnayake, Yukinobu Hoshino.

**Visualization:** Sherin Kularathne, Namal Rathnayake.

**Writing – original draft:** Sherin Kularathne, Namal Rathnayake.

**Writing – review & editing:** Madhawa Herath, Upaka Rathnayake, Yukinobu Hoshino.

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
