## [Decision Letter · Decision Letter 0]

13 Mar 2024

PONE-D-24-05247Economic Factors and Rice Production: A Machine Learning Analysis of Correlations and Predictive ModelsPLOS ONE

Dear Dr. Rathnayake,

Thank you for submitting your manuscript to PLOS ONE. After careful consideration, we feel that it has merit but does not fully meet PLOS ONE’s publication criteria as it currently stands. Therefore, we invite you to submit a revised version of the manuscript that addresses the points raised during the review process.

Please submit your revised manuscript by Apr 27 2024 11:59PM. If you will need more time than this to complete your revisions, please reply to this message or contact the journal office at plosone@plos.org. Please include the following items when submitting your revised manuscript:A rebuttal letter that responds to each point raised by the academic editor and reviewer(s). You should upload this letter as a separate file labeled 'Response to Reviewers'.A marked-up copy of your manuscript that highlights changes made to the original version. You should upload this as a separate file labeled 'Revised Manuscript with Track Changes'.An unmarked version of your revised paper without tracked changes. You should upload this as a separate file labeled 'Manuscript'.

We look forward to receiving your revised manuscript.

Kind regards,

Dr. Sandeep Samantaray

Academic Editor

PLOS ONE

Journal Requirements:

3. In the online submission form, you indicated that [The data will be available upon a format request to the corresponding author.]. 

Additional Editor Comments:

1. Why author took the data set up to 2020; is there any scientific reason- suggestion to add data set upto 2023.

2. Eq 2-5: author must add reference for cite the formulae.

3. Author must add parameter table and pseudo code all proposed algorithm

4. Author must add scatter plot, time series plot, histogram plot for better accuracy of result.

5. Author must add some more discussion in terms of comparison with other study and advantages of proposed model, future scope of the research.

Reviewers' comments:

Reviewer's Responses to Questions

**Comments to the Author**

1. Is the manuscript technically sound, and do the data support the conclusions?

Reviewer #1: Yes

Reviewer #2: Yes

2. Has the statistical analysis been performed appropriately and rigorously? 

Reviewer #1: Yes

Reviewer #2: N/A

3. Have the authors made all data underlying the findings in their manuscript fully available?

Reviewer #1: Yes

Reviewer #2: No

4. Is the manuscript presented in an intelligible fashion and written in standard English?

Reviewer #1: Yes

Reviewer #2: Yes

5. Review Comments to the Author

Reviewer #1: Title: Consider a more specific title that reflects the main findings of your study. For example, "Impact of Economic Indicators on Rice Production: A Machine Learning Approach in Sri Lanka."

Abstract:

Consider including a sentence on the practical implications of the study's findings.

Introduction:

Clearly state the research questions or hypotheses.

Methodology:

Describe the preprocessing steps in more detail, including how missing or inconsistent values were handled.

Results and Discussion:

Discuss the implications of the results in the context of existing literature and theory.

Address any limitations of the study and suggest avenues for future research.

Conclusion:

Summarize the key findings of the study and their implications for agricultural policy and practice.

Highlight the contribution of the study to the existing literature and any practical applications of the findings.

General:

Ensure consistency in terminology and formatting throughout the manuscript.

Reviewer #2: 1. The novelty and contribution of the paper should be clearly stated.

2. What is the contribution of this finding in addressing the challenges that were identified in the introduction?

3. What authors were motivated to conduct this type of study, and why did they select the Sri Lanka as a case study?

4. Plenty of work has already proved the potential of various ML models across the world. What is different being using the applied models in this study?

5. What were the used input parameters?

6. Provide the range of statistical evaluating standards and support them by citing suitable references. The authors can cite https://doi.org/10.1007/978-3-031-12641-3_31; Doi:10.4704/nq.2022.20.14. NQ880113; https://doi.org/10.1063/5.0161098;
https://doi.org/10.1063/5.0132387; DOI: 10.1088/1755-1315/1084/1/012054; DOI: 10.1088/1755-1315/1032/1/012016

7. Likewise, what is the transferability of such results to other locations in terms of impact or usefulness?

8. Include all the figures and tables at their respective place in the text.

9. Authors need to consult with a native English speaker for language improvement.

10. Reveal more practical debate in the discussion section and explain the usefulness and practical implications of this work.

6. PLOS authors have the option to publish the peer review history of their article (what does this mean?). If published, this will include your full peer review and any attached files.

Reviewer #1: No

Reviewer #2: No

---

## [Author Response · Author response to Decision Letter 0]

20 Apr 2024

Responses for Editor's and Reviewer's comments are attached as separate PDF files named 

1. Response_to_Editor_Comments.pdf (Attached as the Cover letter)

2. Response_to_Reviewer_1_Comments.pdf

3. Response_to_Reviewer_2_Comments.pdf

Thank you.

---

## [Editor Report · Decision Letter 1]

2 May 2024

Impact of Economic Indicators on Rice Production: A Machine Learning Approach in Sri Lanka.

PONE-D-24-05247R1

Dear Dr. %Namal Rathnayake%,

We’re pleased to inform you that your manuscript has been judged scientifically suitable for publication and will be formally accepted for publication once it meets all outstanding technical requirements.

Kind regards,

Dr. Sandeep Samantaray

Academic Editor

PLOS ONE
---

## [Editor Report · Acceptance letter]

17 May 2024

PONE-D-24-05247R1 

PLOS ONE

Dear Dr. Rathnayake, 

I'm pleased to inform you that your manuscript has been deemed suitable for publication in PLOS ONE. Congratulations! Your manuscript is now being handed over to our production team.

Kind regards, 

on behalf of

Dr. Sandeep Samantaray 

Academic Editor

PLOS ONE